# Predictors of fear of falling among community-dwelling older adults: Cross-sectional study from Palestine

Manal Badrasawi[1]*, May Hamdan[2], Divya Vanoh[3], Souzan Zidan[4], Tasneem ALsaied[2], Tala B. Muhtaseb[2]

**1** Department of Nutrition and Food Technology, Faculty of Agriculture and Veterinary Medicine, An-Najah National University, Nablus, West Bank, Palestine, **2** Program of Health and Therapeutic Nutrition, Collage of Medicine and Health Sciences, Palestine Polytechnic University, Hebron, Palestine, **3** Dietetics Programme, School of Health Sciences, Health Campus, Universiti Sains Malaysia, Kubang Kerian, Malaysia, **4** Department of Nutrition and Food Technology, Faculty of Agriculture, Hebron University, Hebron, West Bank, Palestine

* m.badrasawi@najah.edu

## Abstract

### Introduction

Fear of falling has serious implications for health and is an important threat to autonomy. The purpose of this cross-sectional study was to investigate risk factors for fear of falling among Palestinian older adults in Hebron district.

### Methods

A cross-sectional study was conducted among Palestinians > 60 years living in Hebron, West Bank. The Falls Efficacy Scale-International was used to predict falls among Palestinian older adults. Moreover, socio-demographic data, medical history, lifestyle habits, body composition, nutritional status, cognitive status (using the Montreal cognitive assessment tool), and functional status (using activities of daily living and instrumental activities of daily living scale), the presence of depressive symptoms (using geriatric depression scale), and physical fitness performance (using senior fitness test) were collected through an interview-based questionnaire. Data were analyzed using univariate and multivariate approach.

### Results

A total of 200 participants were included in the study; 137 (68.5%) females and 63 (31.5%) males. Mean age was 70.5 ± 5.7 years, ranged from 65 to 98 years old. Fear of falling was significantly higher among older adults with advanced aged, living in villages or camps, low educational level, and being married ($p < 0.05$). Functional status (ADL and IADL), physical fitness status (timed up and go), and depression symptoms were significantly related to fear of falling ($p < 0.05$).

### Conclusion

High concern of falling is significantly associated with advanced age, low education level, being married, and living in villages or camps. ADLs were among the factors that had a

**Funding:** The author(s) received no specific funding for this work.

**Competing interests:** the authors have declared that no competing interests exist.

significant relationship with increased fear of falling. Predictors of fear of falling among Palestinian older adults were IADL scores, body fat percentage, rapid gait speed, timed up and go test. Future studies could investigate further correlates of fear of falling among older adults.

## Introduction

There is a global increase in aging population. WHO reports indicates that the world aging population was 1 billion in 2019, and this figure is expected to rise to 2.1 billion by 2050 [1]. In general, this elevation in the relative percentage in comparison to the overall population has great implications both for health services and for older adults [1]. In Palestine, the number of older adults aged more than 60 years in mid-year 2020 was 269,346 people (5% of the population): 177,836 people (6%) in the West Bank and 91,510 people (5%) in Gaza Strip. This percentage of older population is expected to rise even more in the next decade [2]. Evidence shows that older adults' population are at increased risk of falls with consequent serious injury disability, reduced functional capacity and high health cost [3].

An accidental fall has been defined as an unexpected and unintended alteration of posture to the floor, onto a matter, ground, or any other kind of surface, and involves trembling, downfall on people, occasional stooping, lack of equilibrium, and slipping [4, 5]. Falls are major causes of high rates of morbidity and mortality as well as main contributors to precocious nursing home placement and immobility. Accidental injuries deem the fifth leading cause of death among older adults (after cardiovascular disease, cancer, stroke and pulmonary disorders), and falls comprise two-thirds of these deaths [6].

Fear of falling, a psychological symptom known as a "permanent worry about falling that can drive an individual to avert activities that he/she can perform" [7]. Fear of falling is common in 35%–55% among older adults, regardless of whether they have a past fall history [8]. In a systematic review, included six studies conducted among Arabic population, Gulf Cooperation Council Countries, the pooled prevalence of fall was 46.9% ranged from 34% to 57.7% [9]. While in Turkey, the prevalence of falls among older adults aged 80 years and above was 35.4%, and fear of falling was 86.6% [10]. Among Iranian population, the prevalence of falls was 39.7% among 653 older adults [11].

Evaluating all of the three components of fear of falling (i.e., behavioral, physiological, and cognitive) could lead to more precise prediction of fall risk [12]. Self-efficacy beliefs are associated with fear of falling as fear is influenced by beliefs. Hadjistavropoulos and his colleagues (2011) indicated that there is a relationship between the two constructs "fear of falling and self-efficacy", however, they differ from each other [12]. Over the years, many assessment tools have been used to access falls efficacy and fear of falling. The most commonly used tool is the Falls Efficacy Scale–International "FES-I", which consists of 10 items from the original scale and 6 items on social activities (e.g., going to a social event, visiting relatives and friends), and more complex instrumental activities of daily living "IADL" (e.g., ability to use telephone) [13], have been found to be able to predict the risk of falls and to assess the level of falling concern.

Former studies have suggested that fear of falling is related with depression, an elevated risk of falling and low quality of life, and constraints in performing daily activities [14]. Gender, advanced age, physical inactivity, decreased quality of life, social isolation [15], and poor health status [16] were reported to be the causes of fear of falling among older adults individuals.

Gagnon and his colleagues argued that activity avoidance (often developed as a response of fear of falling), raises the risk of fall, functional decline, and pain, since it can result in deconditioning (e.g., immobility, imbalance, lowered function, strength), changes in gait, and poor balance [17]. Moreover, exaggerated activity avoidance can lead to reduced ability to perform activities of daily living (ADLs). Models that make such presumption have been illustrated as fear-avoidance models [e.g., 9].

Evidence also suggested that older adults who have fear of falling show longer double support time, slower gait speed, and shorter stride in comparison to their counterparts [18]. In addition, fear of fall is correlated with gait variability, which is viewed as fall predictor [19].

Given the paucity of studies regarding fear of falling among older adults in the Arab world. Therefore the main aim in the current study was to estimate the prevalence of fear of falling among older adults living in Hebron district, Palestine. We also hypothesized that parameters such as sociodemographic characteristics and lifestyle habits, medical history, body composition and anthropometric measurements, nutritional status, dietary practices, cognitive function, and physical functional status are associated with fear of falling, as previous literature indicated that they are related to aging process and age-related diseases. In the future, we are aiming to conduct longitudinal studies among older adults.

## Methods

### Study design and setting

This cross-sectional study was performed on a representative sample of Palestinian older adults in Hebron district, Palestine. Data collection was done face-to-face throughout a research team consisted of two trained nutritionists to take anthropometric measurements and assess body composition, and one physiotherapist to perform physical function tests. The research team collected the data within six months starting from November 2020 till March 2021. Participants were verbally briefed about the purpose of study, then, pretested-structured questionnaires were distributed to participants upon their signed consent to participate. The collected data included concerns about falling, sociodemographic characteristics, lifestyle habits, medical history, body composition, anthropometric measurements, risk of malnutrition, dietary practices, cognitive function, and physical functional status. The study protocol was approved by the Deanship of Scientific Research Ethical Committee at Palestine Polytechnic University committee. Written informed consent was also obtained from each participant.

**Participants.** *Inclusion and exclusion criteria*. In the current study, participants were included if they aged 65 years old or above, and willing to participate and to provide all the required data. While the exclusion criteria were the presence an acute illness on the days of data collection, having severe hearing problems that prevented communication with the research team, having medical situation that may limit their ability to perform the tests (e.g., participants with severe edema, cachexia, or ascites), having dementia, having current fractures of extremities, do not consenting to participate in the study, and having missing primary data.

*Participants' recruitment*. Participants were recruited from Hebron district, Palestine. A total of 210 participants were invited to join the study and verbally consent to join the study. Only 200 participants were included in the final analysis: 137 (68.5%) females and 63 (31.5%) males. The rest of the participants were excluded mainly due to missing data. The mean age of participants was 70.5 ± 5.7 years, ranged from 65 to 98 years old.

**Sample size and sampling techniques.** Palestine is considered a homogeneous country, as it is a small country divided into three regions (north, center, and south) that share the same traditions and food culture. So we selected Hebron district by stratified sampling method. Then, we randomly selected three different areas from Hebron district. We used G power

software with an alpha of 0.05 (two-sided), CI of 95%, 80% power, and seventeen predictors based on difference between two independent proportions. A minimum of 200 participants was needed to determine the difference between the four groups (no fall concern, low fall concern, moderate and high fall concern) using logistic regression model.

**Measurements.** *Fear of falling.* Many screening instruments e.g., Adapted Falls Efficacy Scale "Adapted FES" [20], Modified Falls Efficacy Scale "MFES" [21], Concern and Fear about falling "CAFlik" [22], Activities-specific balance confidence scale "ABC scale" [23], Survey of Activities and Fear of Falling in the Elderly "SAFFE/SAFE" [24], and Fear of falling subscale "FOF subscale" [25] are available to assess concerns about falling among older adults. One of the most widely accepted and used is Falls Efficacy Scale-International "FES-I", therefore we have used the validated Arabic version of this scale which is commonly used in Arabic countries. The validation study showed that all the items of the scale has a high percentage agreement (from 88 to 93%), and the relative position ranged from 0.01 to 0.06. This scale is rated on a 4-point scale for each activity, where: 1 = not at all concerned, 2 = somewhat concerned, 3 = fairly concerned and 4 = very concerned. The cumulative score ranges from 16–64. (no concern about falling) to 64 (severe concern about falling) [26]. According to FES-I protocol, older adults were also asked if they had a history of fall within the past six months.

*Demographics and lifestyle habits.* Questions regarding demographic data, including age, gender, marital status, area of living, living status, and economical status, were asked for each participant.

Data about lifestyle habits (e.g., smoking, the type of smoking whether cigarette or pipe duration of smoking, duration of smoking cessation, having either sleeping problems or sleep apnea, duration of sleeping, and physical activity level) were elicited from the participants. Participants were classified into two categories according to their sleeping duration and the presence of sleeping problems. Participants were classified within good quality sleepers' group if their sleeping duration ranges from six to eight hours and didn't sleeping problems (e.g., inability to get a sleep, insomnia), while the rest of participants were categorized as poor-quality sleepers.

Physical activity level was measured using the Arabic short -form of international physical activity questionnaire (IPAQ), which has a high reliability ($\alpha < 0.80$). According to IPAQ scoring protocol, the metabolic equivalent. minutes per week (MET.min/week) was calculated for each of walking, moderate- and vigorous-intensity activities [27]. A score of $\leq 3$ MET indicates low level of physical activity, a score of 3–6 MET indicates moderate level of physical activity, and a score of $\geq 6$ MET indicates a high level of physical activity [28].

*Medical history.* In this section, health status was determined by dichotomized questions (yes/no) about the presence of diseases (hypertension, hypercholesterolemia, diabetes, stroke, osteoarthritis, heart disease, glaucoma, renal disease, asthma, chronic obstructive pulmonary disease, gout, hip fracture, constipation, gastric ulcer, vision problems, urinary problems). The participants were also asked to report if they undergo a surgery (how many times? when, and type of surgery), if they suffer from cancer (how many times? type of cancer, and the type of cancer treatment "i.e., surgery, biological, radiology, chemotherapy", and if they suffer from eating problems "i.e., loss of appetite, chewing problems, or gastric problems such as nausea, vomiting, and diarrhea").

*Body composition and anthropometric measurements.* In this section, older adults' anthropometric indices (e.g., weight, height, waist circumference, hip circumference, calf circumference and mid upper arm circumference "MUAC") were measured according to the standard method described by Lee &Nieman [29]. Moreover, the body composition data was analyzed using a portable machine named InBody 120. Data collection team has requested from the participants to take off their socks and their shoes, and to put any metallic accessories (e.g., rings,

watch, bracelet) on the table. Then, the participants were asked to step on the machine with their bared feet and catch the hand device with their hands. Fat mass, fat percentage, fat free mass, muscle mass, skeletal muscle mass, body mass index, and basal metabolic rate were extracted from the machine output. Body mass index was classified according to WHO cut off points [30].

*Nutritional status*. We have performed clinical assessment of participants' nutritional status by using a validated screening tool named Mini Nutritional Assessment-Short form-Arabic Version (MNA-SF-A). A previous study showed that the original scale is valid with sensitivity equal to 97.9% and specificity of 100% [31]. This tool is composed of 6 questions including some anthropometric measurements (e.g., BMI and calf circumference). The total score ranges from 0 to 14. A score of $\geq$ 12 indicates absence of malnutrition, a score of 8–11 points indicates that older adults is at risk of malnutrition, and a score of $\leq$ 7 points indicates that older adults is malnourished [32].

*Dietary practices and gastric problems*. This part of the questionnaire was designed to assess participants' dietary practices. The participants were asked the following dietary practices -related questions; the number of main meals and snacks per day, participants' appetite, duration of overnight fasting, and the amount of consumed meal compared to usual.

*Depressive symptoms and cognitive function*. A validated Arabic version of the short geriatric depression scale (GDS-15), which has a high sensitivity (83%) and specificity (91%), was used to detect depression among older adults. The scale is consisted of 15 dichotomized questions (yes/no) concerning participants mood. One point is given either to answer "yes" or to the answer "no" depending on the question, then, the cumulative score is rated on a scoring grid. A score of $<$ 5 indicates absence of depression and a score of $\geq$ 5 indicates a high risk of a depressive disorder [33]. Moreover, a validated Arabic version of Montreal cognitive assessment–basic was used to detect mild cognitive impairment [34]. This tool has a sensitivity of 92.3% and specificity of 85.7% [34]. The Montreal cognitive assessment–basic assesses the following cognitive domains (language, orientation, executive function, simple mathematical calculations, memory, conceptual thinking, visual perception, attention and concentration). The questionnaire is scored on 30 points: scores $\geq$ 26 are considered normal, scores $<$ 26 indicate mild cognitive impairment [34].

*Physical functional status*. Functional status among elderly was assessed by using a validated Arabic versions of both ADLs and IADL. The ADLs, which has a sensitivity of 38% and a specificity of 80%, was assessed using the Katz index scale [35]; and the IADL, which has a sensitivity of 62% and a specificity of 80% at the lowest cut-off point for the diagnosis of cognitive impairment, was assessed using the Lawton scale [36]. Physical fitness was performed using senior fitness tests. The following tests were selected to assess participants' physical fitness: handgrip for upper body strength, which is a valid predictive measure for age-related disorders [37]; 30-seconds chair stand test as a reliable and valid indicator for lower body strength [38]; back scratch test for upper body flexibility [39]; set and reach for lower body flexibility [39]; 8-ft time up and go as a valid test for balance [39]; 2-minute step test to assess cardiovascular fitness and endurance [39], and gait speed for pace assessment [39].

## Statistical analysis

SPSS version 26.0 was used for analyzing the data. Normality testing was performed using the histogram. Descriptive statistics was used for presenting participant's demographic characteristics. Independent-t-test was employed to identify the mean differences of nutritional status, cognitive function, and physical function between gender. Chi-Square test was used to determine the association between categorical variables such as gender, marital status, smoking,

medical problems, nutritional status with gender and fear of falling. Kruskal-Wallis test was employed to investigate the median differences between the categorical variable (fear of falling) and numerical data (age, education years, physical fitness test, cognitive function, functional status, depressive symptoms and sleep quality). On the other hand, multivariate analysis was conducted using the ordinal logistic regression with falls concern as dependent variable. The independent variables chosen for the regression model were the ones significant from the univariate analysis and the those identified as important factors related to fear of falling from previous studies. Multicollinearity was determined using the VIF value >10 in linear regression with dummy coded dependent variable. In the ordinal logistic regression model, the dependent variable is coded as 0: Low falls concern, 1: moderate falls concern, 2: high falls concern. The ordinal regression model was adjusted for age, education years, gender, marital status, smoking status, body mass index. Significant level was set at $p < 0.05$.

## Results

### Participants' characteristics

Participants' demographic characteristics, nutritional status, cognitive, and physical functional status are presented in Tables 1 and 2, respectively. In terms of dietary habits, we have found that the mean number of consumed meals per day was 2.4±0.60 meals per day, the mean number of snacks consumed daily was 1.5±0.69 meals daily, and the mean hours of overnight fasting was 7.9 ±1.42 hours.

The analysis of IPAQ survey reveals that only 6.0% of participants (n = 6) showed a high level of physical activity, 31.0% of them (n = 62) showed a moderate level of physical activity and a relatively a high percentage of them 63.0% (n = 126) showed a low level of physical activity. Moreover, among 200 participants, 68 (34.0%) were classified within poor sleeping quality group, while the rest of participants were classified within good sleeping quality group.

The results of the medical history revealed considerable prevalence of cardiovascular diseases among the participants; hypertension was reported among 51.5% of the participants,

**Table 1. Participants' demographic characteristics presented in numbers (n) and percentages (%).**

| Variables | | Total (n = 200) | |
|---|---|---|---|
| | | n | % |
| Gender | Males | 63 | 31.5 |
| | Females | 137 | 68.5 |
| Marital status | Single | 9 | 4.5 |
| | Married | 146 | 73.0 |
| | Widow | 45 | 22.5 |
| Living status | Alone | 33 | 16.5 |
| | Spouse only | 93 | 46.5 |
| | Family members | 74 | 37.0 |
| Place of residence | City | 33 | 16.5 |
| | Village | 166 | 83.0 |
| | Camp | 1 | 0.5 |
| Current employment status | Working | 36 | 18.0 |
| | Not working | 164 | 82.0 |
| Monthly income | <1300 NIS | 114 | 57.0 |
| | ≥ 1300 NIS | 86 | 43.0 |

NIS: New Israeli Shekel.

**Table 2. Participants' nutritional status, cognitive and physical function profile according to gender.**

| Variables | | Males (n = 63) Mean ± SD | Females (n = 137) Mean ± SD | p-value |
|---|---|---|---|---|
| Nutritional status | BMI | 30.3 ± 5.2 | 33.6 ± 6.1 | 0.000* |
| | MUAC | 32.7 ± 3.4 | 32.1 ± 3.6 | 0.221 |
| | CC | 37.0 ± 5.2 | 37.8 ± 6.0 | 0.573 |
| | Waist: hip ratio | 1.0 ± 0.1 | 0.9 ± 0.1 | 0.000* |
| | Body fat mass [kg] | 32.4 ± 13.6 | 29.6 ± 11.7 | 0.152 |
| | Body fat percentage [%] | 33.1 ± 8.8 | 31.3 ± 7.5 | 0.052 |
| | FFM [kg] | 53.5 ± 9.5 | 51.6 ± 8.6 | 0.248 |
| | SMM [kg] | 44.4 ± 14.2 | 41.2 ± 13.2 | 0.108 |
| | BMR [kcalories] | 1599.0 ± 227.1 | 1579.1 ± 229.5 | 0.659 |
| | MNA | 12.7 ± 1.7 | 12.4 ± 2.1 | 0.822 |
| Cognitive function | MOCA | 25.9 ± 4.5 | 26.1 ± 3.5 | 0.900 |
| Depression symptoms | GDS | 2.8 ± 2.7 | 4.0 ± 3.1 | 0.008* |
| Functional status | ADL | 5.7 ± 0.7 | 5.4 ± 1.1 | 0.159 |
| | IADL | 5.1 ± 1.2 | 6.3 ± 1.9 | 0.000* |
| Physical fitness | 2 min step test | 65.6 ± 27.2 | 53.4 ± 24.8 | 0.004* |
| | BST | 19.7 ± 9.3 | 17.9 ± 11.1 | 0.186 |
| | CSR | 4.6 ± 7.6 | 3.2 ± 5.8 | 0.272 |
| | CST | 12.1 ± 4.0 | 11.7 ± 3.8 | 0.386 |
| | Gate speed | 8.7 ± 1.7 | 9.0 ± 2.4 | 0.598 |
| | Hand grip | 30.5 ± 9.9 | 20.0 ± 5.7 | 0.000* |
| | RP | 5.6 ± 1.4 | 5.7 ± 1.7 | 0.810 |
| | TUG | 12.4 ± 2.6 | 13.6 ± 3.4 | 0.026* |

*Significant p<0.05 using independent t-test.

SD: standard deviation; BMI: body mass index, MUAC: mid upper arm circumference, CC: calf circumference; FFM: fat free mass; SMM: skeletal muscle mass; BMR: basal metabolic rate; MNA: mini nutritional assessment; MOCA: Montreal cognitive assessment; ADL: activity of daily living; IADL: instrumental activity of daily living GDS: Geriatric depression scale, 2min.st: 2 minutes step test, BST: back scratch test; CSR: chair set and reach test; CST: chair stand test; RP: rapid pace; TUG: timed up and go.

followed by diabetes mellitus was reported among 43.0% of the participants. Whereas renal failure, and chronic obstructive pulmonary disease were the least common among our participants by 3.0%, and 3.0% respectively.

## Fear of falling

Based on the total scores obtained in Falls Efficacy Scale-International, the participants were divided into high falls concern, moderate falls concern, low falls concern, and no falls concern groups. In the high falls concerns group, there were 24.0% (n = 48) participants. In the moderate concerns group, there were 25.0% (n = 50) participants and in the low concern group, there 22.5% (n = 45) of participants. While 28.5% (n = 57) participants were in the low falls concern group (Fig 1) The observed risk factors associated with fear of falling are listed in Tables 3–6.

## Sociodemographic characteristics according to fear of falling categories

Table 3 shows the relationship between participants' sociodemographic characteristics and fear of falling. And as you can see, the risk of falls demonstrated significantly higher concerns

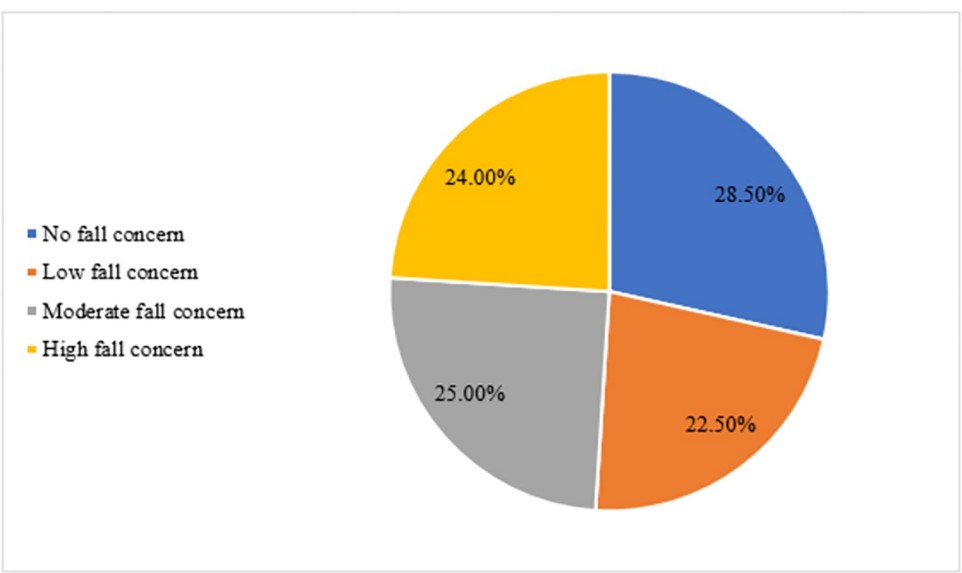

**Fig 1. Fear of falling.**

about falling among individuals who were older (74.15±7.68), less educated (7.79±4.83), married (60.4%), and residing in village or camps (91.7%) ($p <0.05$).

Findings from Table 4 shows that older adults with high falls concern have lower body fat percentage (28.40±8.31), higher fat free mass (55.59±8.50), compared to older adults with no risk concern ($p <0.05$). Moreover, we noticed that skeletal muscle mass, body mass index, and basal metabolic rate were significantly the highest among older women with high falls concern ($p <0.05$). Furthermore, older women with high falls concern had significantly lower MUAC, CC, and waist to hip ratio compared to older women with no falls concern ($p<0.05$). Furthermore, the number of snacks eaten per day were significantly higher among older women with high falls concern compared to women with no falls concern. The analysis also shows that malnutrition was significantly more prevalent among older adults with high falls concern compared to their counterparts (no falls concern, low falls concern, and high falls concern) ($p <0.05$), however, the difference was not significant after the analysis was done separately for males and females.

As shown in Table 5, older men with high concern about falling had significantly lower physical performance in the following tests; gate test, hand grip, rapid pace, and timed up and go compared to older men with no falls concern. Our analysis also reveals that older women had significantly a worse performance in all physical function tests in comparison with older women with no falls concern. Moreover, IADL and ADL scores were also significantly low among older adults with high falls concern as compared to their counterparts ($p<0.001$). Besides that, we have found that older adults with high falls concern had significantly the lowest MOCA score and the highest GDS score indicating poor cognitive function and higher depressive symptoms as compared to other groups ($p<0.05$).

Among the predictors of falls concern were IADL scores, body fat percentage, rapid gait speed, chair stand test, timed up and go test. One unit increase in IADL score lead to a decrease in 0.631 in the ordered log odds of being in the high falls category. Older people with good ability to perform IADL have lower risk of being in the high fear of falling category. Besides that, for each unit increase in the rapid gait speed score will increase 1.548 in the ordered log odds of being in the high falls category which indicated that poor performance in

**Table 3. Sociodemographic characteristics according to fear of falling.**

| Sociodemographic characteristics | Fear of falling | | | | p-value |
|---|---|---|---|---|---|
| | No falls concern | Low falls concern | Moderate falls concern | High falls concern | |
| Age [years, mean ± SD] | 68.75±4.20 | 68.82±4.05 | 70.30±4.79 | 74.15±7.68 | 0.000* |
| Years of education [years, mean ± SD] | 10.12±3.60 | 9.58±4.04 | 9.58±4.10 | 7.79±4.83 | 0.042* |
| Gender | | | | | 0.275 |
| Men [n (%)] | 23 (40.4) | 15 (33.3) | 12 (24.0) | 13 (27.1) | |
| Women [n (%)] | 34 (59.6) | 30 (66.7) | 38 (76.0) | 35 (72.9) | |
| Marital status | | | | | 0.015* |
| Single [n (%)] | 3 (5.3) | 4 (8.9) | 2 (4.0) | 0 (0.0) | |
| Married [n (%)] | 47 (82.5) | 34 (75.6) | 36 (72.0) | 29 (60.4) | |
| Widow [n (%)] | 7 (12.3) | 7 (15.6) | 12 (24.0) | 19 (39.6) | |
| Living status | | | | | 0.329 |
| Alone [n (%)] | 5 (8.8) | 7 (15.6) | 10 (20.0) | 11 (22.9) | |
| Spouse only [n (%)] | 32 (56.1) | 20 (44.4) | 24 (48.0) | 17 (35.4) | |
| Family members [n (%)] | 20 (35.1) | 18 (40.0) | 16 (32.0) | 20 (41.7) | |
| Place of residence | | | | | 0.001* |
| City [n (%)] | 19 (33.3) | 4 (8.9) | 6 (12.0) | 4 (8.3) | |
| Village/ Camp [n (%)] | 38 (66.7) | 41 (91.1) | 44 (88.0) | 44 (91.7) | |
| Current employment status | | | | | 0.043* |
| No work [n (%)] | 42 (73.7) | 35 (77.8) | 47 (94.0) | 40 (83.3) | |
| Have work [n (%)] | 15 (26.3) | 10 (22.2) | 3 (6.0) | 8 (16.7) | |
| Monthly income | | | | | 0.275 |
| <1300 NIS [n (%)] | 38 (66.7) | 25 (55.6) | 24 (48.0) | 27 (56.2) | |
| ≥ 1300 NIS [n (%)] | 19 (33.3) | 20 (44.4) | 26 (52.0) | 21 (43.8) | |
| Smoking status | | | | | 0.076 |
| Smoker [n (%)] | 46 (80.7) | 36 (80.0) | 38 (76.0) | 43 (89.6) | |
| Non-smoker [n (%)] | 10 (17.5) | 3 (6.7) | 6 (12.0) | 2 (4.2) | |
| Former smoker [n (%)] | 1 (1.8) | 6 (13.3) | 6 (12.0) | 3 (6.2) | |
| Physical activity level | | | | | 0.066 |
| Low physical activity [n (%)] | 30 (52.6) | 25 (55.6) | 33 (66.0) | 38 (79.2) | |
| Moderate physical activity [n (%)] | 21 (36.8) | 16 (35.6) | 16 (32.0) | 9 (18.8) | |
| High physical activity [n (%)] | 6 (10.5) | 4 (8.9) | 1 (2.0) | 1 (2.1) | |

Data are presented as *n* (%) or mean ± SD

* *p* < 0.05.

Pearson chi-square test is employed for categorical variables and Kruskal-Wallis test for continuous variables.

SD: standard deviation; NIS: new Israeli shekel.

the rapid gait speed test increases fear of falling. On the other hand, individuals with higher body fat percentage have higher risk of being in the high falls concern group (OR: 0.881; 95% CI: -0.227; -0.027). Minimal body fat percentage can be a risk factor of fear of falling. As for chair stand test, the higher count in the test indicated a 0.858 decrease in the logs odds of being in the high fear of falling group (Table 6).

## Discussion

In the current study, we targeted to estimate the prevalence of fear of falling among a sample of Palestinian older adults and investigate possible factors that might be related to developing a

 

**Table 4. Nutritional status according to fear of falling.**

| | Males | | | | | Females | | | | |
|---|---|---|---|---|---|---|---|---|---|---|
| | Fear of falling | | | | | Fear of falling | | | | |
| Variables | No falls concern | Low falls concern | Moderate falls concern | High falls concern | *p*-value | No falls concern | Low falls concern | Moderate falls concern | High falls concern | *p*-value |
| MUAC | 32.9±3.0 | 33.6±4.0 | 31.3±2.1 | 32.7±4.0 | 0.278 | 30.9±3.1 | 31.7±3.0 | 33.2±4.2 | 32.6±3.7 | 0.041* |
| CC | 36.8±5.7 | 38.0±6.1 | 36.3±4.0 | 36.7±4.5 | 0.636 | 35.6±3.5 | 37.6±4.4 | 39.3±8.6 | 38.4±5.1 | 0.021* |
| Waist to hip ratio | 1.02±0.2 | 0.96±0.1 | 0.98±0.0 | 0.94±0.0 | 0.070 | 0.92±0.1 | 0.85±0.1 | 0.90±0.2 | 0.94±0.1 | 0.001* |
| Body fat mass | 38.6±12.5 | 30.5±16.1 | 32.5±10.2 | 23.5±10.5 | 0.015* | 31.2±8.7 | 27.0±10.7 | 30.5±12.3 | 29.4±14.3 | 0.312 |
| Body fat percentage | 37.4±6.7 | 31.3±10.7 | 34.5±5.8 | 26.3±8.0 | 0.002* | 35.1±5.0 | 28.9±7.0 | 31.7±7.7 | 29.2±8.4 | 0.001* |
| FFM | 52.0±11.4 | 56.1±8.3 | 49.1±4.9 | 57.2±8.5 | 0.016* | 45.6±3.6 | 51.5±9.3 | 53.8±8.7 | 55.0±8.5 | 0.000* |
| SMM | 43.4±13.7 | 47.3±15.0 | 37.9±8.7 | 49.1±17.1 | 0.136 | 33.1±6.9 | 41.4±11.2 | 44.1±15.3 | 45.4±14.2 | 0.000* |
| BMI | 31.1±6.1 | 31.0±5.7 | 29.5±3.4 | 28.7±4.2 | 0.675 | 30.4±4.0 | 33.00±5.4 | 34.5±6.1 | 36.2±6.9 | 0.000* |
| BMR | 1540.7 ±215.9 | 1697.3 ±239.0 | 1510.7±117.1 | 1670.7±264.4 | 0.077 | 1411.9 ±134.8 | 1611.4 ±203.9 | 1628.4±256.0 | 1660.5±218.8 | 0.000* |
| Dietary habits | | | | | | | | | | |
| Number of meals/day | 2.48±0.7 | 2.53±0.5 | 2.42±0.5 | 2.46±0.7 | 0.944 | 2.44±0.6 | 2.43±0.6 | 2.42±0.6 | 2.34±0.6 | 0.867 |
| Number of snacks/day | 1.48±0.7 | 1.27±0.9 | 1.58±0.6 | 1.85±0.6 | 0.213 | 1.26±0.5 | 1.60±0.6 | 1.68±0.8 | 1.63±0.6 | 0.008* |
| Overnight fasting | 7.87±1.4 | 7.20±1.1 | 7.6±1.8 | 8.15±1.5 | 0.325 | 8.09±1.1 | 7.43±1.3 | 8.03±1.5 | 8.31±1.6 | 0.114 |
| MNA | | | | | 0.488 | | | | | 0.156 |
| Malnourished | 0(0.0) | 0(0.0) | 0(0.0) | 1(7.7) | | 1(2.9) | 0(0.0) | 1(2.6) | 5(14.3) | |
| At risk of malnutrition | 5(21.7) | 2(13.3) | 3(25.0) | 4(30.8) | | 6(17.6) | 6(20.0) | 9(23.7) | 8(22.9) | |
| Well-nourished | 18(78.3) | 13(86.7) | 9(75.0) | 8(61.5) | | 27(79.4) | 24(80.0) | 28(73.7) | 22(62.9) | |

Data are presented as *n* (%) or mean ± SD

* *p* < 0.05.

Pearson chi-square test is employed for categorical variables and Kruskal-Wallis test for continuous variables.

SD: standard deviation; BMI: body mass index, MUAC: mid upper arm circumference, CC: calf circumference; FFM: fat free mass; SMM: skeletal muscle mass; BMR: basal metabolic rate; MNA: mini nutritional assessment.

fear of falling. Fear of falling is considered one of the most remarkable clinical symptoms affecting older adults individuals and a prevalent health issue facing them later in life [40].

The present study findings showed that fear of falling is a prevalent concern among older adults individuals living in Hebron district and common among them by 49.0%. The prevalence of fear of falling in our sample was generally lower than that reported in previous studies including; Egypt (64.6%) [41], Japan (60.0%) [42], and Thailand (95.6%) [43]. These variations in fear of falling prevalence maybe explicated as the current study included sample from urban and rural regions.

Advancing age was significantly associated to fear of falling among older adults. This result is in parallel line with studies conducted by Saleh et al. (2018) [41], Hoang et al. (2017) [44], and Lawson (2013) [45] which reported that increased age was associated with the risk of fear of falling. Our finding can explained as aging is accompanied by numerous deteriorating changes including immobility and reduced functional capacity, physical frailty, cardiac and neuromuscular homeostatic mechanisms, which may either place older adults at fear of falling or at risk for fall [46]. Moreover, phobias are prevalent among older adults; as they become more concern about their health and fear specific situations and accidents, especially falls [47]. On the otherhand, it was stated by Mann et al. (2006) that aging is not a precursor to fear of

**Table 5. Physical and cognitive function characteristics according to fear of falling.**

| | | Males | | | | | Females | | | |
|---|---|---|---|---|---|---|---|---|---|---|
| | | Fear of falling | | | | *p*-value | Fear of falling | | | | *p*-value |
| | | No falls concern | Low falls concern | Moderate falls concern | High falls concern | | No falls concern | Low falls concern | Moderates falls concern | High falls concern | |
| Physical fitness | 2min.st | 69.7±28.6 | 76.9±22.4 | 50.1±19.5 | 59.8±30.3 | 0.071 | 52.0±21.6 | 64.7±25.0 | 53.8±23.7 | 44.5±26.0 | 0.016* |
| | BST | 18.3±9.2 | 17.1±8.9 | 20.8±6.4 | 24.4±11.1 | 0.203 | 14.2±8.6 | 15.2±10.6 | 15.7±9.9 | 26.2±11.3 | 0.000* |
| | CSR | 5.1±9.0 | 3.0±5.6 | 5.8±9.6 | 4.5±4.8 | 0.691 | 1.7±4.6 | 3.2±5.7 | 2.8±5.8 | 4.9±6.7 | 0.018* |
| | CST | 12.1±3.9 | 13.6±3.7 | 9.5±4.1 | 12.8±3.8 | 0.058 | 11.4±3.3 | 14.3±3.5 | 11.7±3.5 | 9.7±3.8 | 0.000* |
| | Gait test | 8.0±1.6 | 8.5±1.4 | 8.9±1.6 | 10.0±1.7 | 0.006* | 7.5±1.7 | 8.9±1.4 | 8.9±2.3 | 10.8±2.6 | 0.000* |
| | Hand grip | 33.6±9.2 | 33.6±10.4 | 26.5±7.9 | 25.0±9.5 | 0.017* | 21.9±5.7 | 21.0±5.6 | 20.0±3.9 | 17.3±6.8 | 0.009* |
| | RP | 5.3±1.2 | 4.7±01.0 | 6.5±1.4 | 6.3±1.6 | 0.003* | 5.0±1.3 | 5.2±1.1 | 5.6±1.5 | 7.0±1.8 | 0.000* |
| | TUG | 11.3±2.4 | 12.3±2.2 | 12.3±2.5 | 14.7±2.5 | 0.004* | 11.6±2.7 | 12.7±2.2 | 13.6±3.3 | 16.3±3.2 | 0.000* |
| Functional status | ADL | 5.9±0.2 | 6.0±0.0 | 5.6±0.7 | 5.0±1.1 | 0.000* | 6.0±0.0 | 5.9±0.6 | 5.7±0.6 | 4.2±1.5 | 0.000* |
| | IADL | 5.5±0.9 | 5.5±1.0 | 4.6±1.1 | 4.5±1.6 | 0.024* | 7.3±0.7 | 7.2±0.9 | 6.3±1.7 | 4.3±2.1 | 0.000* |
| Cognitive function | MOCA | 27.9±3.2 | 26.3±2.2 | 25.2±2.0 | 22.6±7.5 | 0.000* | 28.4±2.3 | 26.7±1.9 | 26.3±2.6 | 23.2±4.3 | 0.000* |
| Depression symptoms | GDS score | 1.4±1.7 | 2.7±1.8 | 3.3±2.3 | 5.1±3.7 | 0.002* | 2.0±1.7 | 3.7±3.4 | 4.0±2.1 | 6.2±3.4 | 0.000* |
| Sleeping quality | Poor | 9(39.1) | 5(33.3) | 6(50.0) | 2(15.4) | 0.311 | 14(41.2) | 8(26.7) | 15(39.5) | 9(25.7) | 0.379 |
| | Good | 14(60.9) | 10(66.7) | 6(50.0) | 11(84.6) | | 20(58.8) | 22(73.3) | 23(60.5) | 26(74.3) | |

Data are presented as *n* (%) or mean ± SD

* *p*-value < 0.05.

Pearson chi-square test is employed for categorical variables and Kruskal-Wallis test for continuous variables.

SD: standard deviation; 2min.st: 2 minutes step test, BST: back scratch test; CSR: chair set and reach test; CST: chair stand test; RP: rapid pace; TUG: timed up and go; ADL: Activities of Daily Living; IADL: Instrumental Activity of Daily Living; MOCA: Montreal Cognitive Assessment Score; GDS: Geriatric Depression Scale.

falling and there are various factors can result in fear of falling such as psychological and physical characteristics [48].

We also noticed that older adults with lower levels of education reported more fear of falling. This result is supported by a former study conducted by Saleh et al. (2018) which found that illiterate older adults experienced high concerns about falling compared to their counterparts [41]. In Minnesota, it was reported by Bagley et al. (2017) that there is a significant relationship between years of education and fear of falling [49]. Other demographic characteristics of older adults that had a significant correlation with the fear of falling were employment status, place of residence, and marital status.

The negative association between fear of falling and declining in functional ability can be confirmed by the findings of the current study. As we noticed that the more dependent older adults were, the greater their fear of falling. It is rational that growing need for aid with activities of daily living may make older adults to feel less safe about their physical abilities, and thus increasing their fear of falling. This finding is consistent with former studies [44, 50], which found a correlation between ADLs score and fear of falling.

Besides that, our study revealed that depressed older adults are more likely to suffer from fear of falling. This is in accordance with many former studies which reported that depression is one of the predictors of fear of falling [44, 50–52]. Legters 2002 stated that 'depression decreased the performance of automatic daily behaviors and in turn decreased the positive reinforcement that comes to a person' [53]. Decreased positive reinforcement triggers a chain of events that may result in reduced involvement in delightful activities, raised necessity for

**Table 6. Predictors of fear of falling.**

| Parameters | Estimate | Standard Error | Odd Ratio | 95% CI | p-value |
|---|---|---|---|---|---|
| Muscle mass | 0.099 | 0.062 | 1.104 | -0.021; 0.220 | 0.107 |
| Body fat percentage | -0.127 | 0.051 | 0.881 | -0.227; -0.027 | 0.013* |
| GDS score | 0.243 | 0.244 | 1.275 | -0.002;0.278 | 0.053 |
| IADL | -0.460 | 0.150 | 0.631 | -0.753;-0.166 | 0.002* |
| Usual gait speed | -0.217 | 0.150 | 0.805 | -0.511;0.076 | 0.146 |
| Rapid gait speed | 0.437 | 0.188 | 1.548 | 0.069;0.805 | 0.020* |
| Two-minute step test | 0.012 | 0.009 | 1.012 | -0.006;0.029 | 0.199 |
| Chair stand test | -0.153 | 0.062 | 0.858 | -0.274;-0.032 | 0.013* |
| Chair sit and reach | 0.007 | 0.028 | 1.007 | -0.048;0.062 | 0.814 |
| Back Scratch Test | -0.022 | 0.019 | 0.978 | -0.058;0.014 | 0.237 |
| Calf circumference | -0.035 | 0.040 | 0.966 | -0.112;0.043 | 0.385 |
| Snacking frequency | 0.243 | 0.244 | 1.275 | -0.236;0.722 | 0.320 |
| Night fasting | -0.053 | 0.113 | 0.948 | -0.275;0.168 | 0.638 |
| MOCA score | -0.132 | 0.074 | 0.876 | -0.307;-0.032 | 0.074 |
| TUG score | 0.144 | 0.068 | 1.102 | 0.011;0.278 | 0.034* |
| Hand grip strength | -0.017 | 0.028 | 0.983 | -0.072;0.038 | 0.545 |
| MNA | | | | | |
| Malnourished | 0.207 | 1.256 | 1.230 | -2.254;2.668 | 0.869 |
| At risk | -0.279 | 0.403 | 0.757 | -1.069;0.510 | 0.488 |
| Well-nourished | Ref | | | | |

Model adjusted for age, education years, gender, income, living area; marital status, smoking status, body mass index; physical activity. Dependent variable: (1: Low falls concern, 2: somewhat falls concern, 3: fairly concern, 4: high falls concern); Abbreviation: IADL: instrumental activities of daily living; GDS: geriatric depression symptoms; MOCA: Montreal Cognitive Assessment; MNA: Mini Nutrition Assessment; 95% CI: 95% Confidence Interval.

*No multicollinearity exist between the independent variables.

help, raised concentrate on the individual's self, and increased need for assistance, unsurprisingly, raised fear of falling [44]. Furthermore, depression is usually accompanied by reduced energy and fatigue, which may make individuals less convinced in their physical capabilities and thus more concerned about falling [54].

Also consistent with previous literature [44], our study noticed a significant correlation between participants' TUG test times and fear of falling among older adults. Better balance and gait status is distinctly related to lower concern about falling for the current sample. Bandura (1999) stated that people depend partially on their physical status in judging their overall self-efficacy and abilities. Age-related alterations in gait and balance status, as well as clinically observed gait and balance problems, which mirror the influences of different disease processes on various components of gait and balance status, may result in increased fear of falling among older adults [55].

Yardley and Smith reported that fear of falling is correlated with poorer health and balance, indicating a negative effect of fear of falling to balance [56]. Shahar and his colleagues, for their parts, revealed that good nutritional status improves both functional abilities and balance abilities among older adults [57]. Moreover, there is evidence that poor nutritional status results in muscle weakness and low muscle mass, which in turn leads to impaired mobility and the necessity for using ambulatory aids like walker [58]. However, studies in relation to nutritional status and fear of falling are still scarce.

According to the current study, the number of snacks eaten per day were significantly higher among older women with high concern about falling. The analysis also shows that

malnutrition was significantly more prevalent among older adults with high falls concern compared to their counterparts (no falls concern, low falls concern, and high falls concern). The association between nutritional status has not been examined before among Palestinian older adults.

Interestingly, we found that older adults with high falls concern have higher body fat percentage, lower fat free mass, compared to older adults with no risk concern. We also noticed that the body mass index, and basal metabolic rate were significantly the highest among older women with high falls concern. Furthermore, older women with high falls concern had significantly lower MUAC, CC, and waist to hip ratio compared to older women with no falls concern.

Findings of the current study must be considered in the framework of its design limitations. Firstly; the major limitation of the current study resides in its design. Being cross-sectional, it's impossible to derive a cause-effect relationship. Secondly, the data collected were self-reported. This may lead to misreporting and recall bias because of the nature of the study and the older age of the participants. Thirdly, the study was limited to only one region in Palestine and doesn't exemplify all older adults' category in Palestine. Nonetheless, our study is the first of its kind to explore the prevalence of fear of falling among older adults in Palestine and its' associated factors. Future studies should focus on clarifying the causal relationship, and assess nutritional and social status for a better understanding of fear of falling among older adults.

## Conclusion

Fear of falling is a prevalent concern of older people and has unfavorable consequences. The findings from this research showed that advanced age, low education level, being married, and living in villages or camps were significantly associated with high concern about falling among Palestinian older adults. ADLs were among the factors that had a significant relationship with increased fear of falling. Predictors of fear of falling among Palestinian older adults were IADL scores, body fat percentage, rapid gait speed, timed up and go test. Future research could replicate this study in other settings, such as long-term care facilities, and in other cities.

## Supporting information

**S1 Data. Final data fall risk.**
(XLS)

## Acknowledgments

We would like to acknowledge the students who helped the researchers in the data collection. We would like to express our gratitude to the Palestinian older adults who agreed to participate in this study. Thanks are also to all co-researchers and fieldworkers involved in this study.

## Author Contributions

**Conceptualization:** Manal Badrasawi, May Hamdan, Tasneem ALsaied, Tala B. Muhtaseb.

**Data curation:** Divya Vanoh.

**Formal analysis:** Divya Vanoh, Souzan Zidan, Tasneem ALsaied, Tala B. Muhtaseb.

**Investigation:** Manal Badrasawi, May Hamdan, Tala B. Muhtaseb.

**Methodology:** Manal Badrasawi, May Hamdan, Divya Vanoh, Tasneem ALsaied, Tala B. Muhtaseb.

**Project administration:** May Hamdan.

**Writing – original draft:** Divya Vanoh, Souzan Zidan, Tasneem ALsaied, Tala B. Muhtaseb.

**Writing – review & editing:** Manal Badrasawi, May Hamdan.

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
