## [Decision Letter · Decision Letter 0]

19 May 2022

PONE-D-21-23236Predictors of fall risk among community-dwelling older adults: cross-sectional study from Palestine.PLOS ONE

Dear Dr. Badrasawi,

Thank you for submitting your manuscript to PLOS ONE. After careful consideration, we feel that it has merit but does not fully meet PLOS ONE’s publication criteria as it currently stands. Therefore, we invite you to submit a revised version of the manuscript that addresses the points raised during the review process.

The manuscript has been evaluated by three reviewers, and their comments are available below.

The reviewers have raised a number of major concerns that need attention. They request additional information on methodological aspects of the study, revisions to the statistical analyses and they question the internal and external validity of the results reported.

Could you please revise the manuscript to carefully address the concerns raised?

We look forward to receiving your revised manuscript.

Kind regards,

Sebastian Shepherd

Staff Editor

PLOS ONE

Journal Requirements:

4. Please upload a new copy of Figure 2 as the detail is not clear. Please follow the link for more information: https://blogs.plos.org/plos/2019/06/looking-good-tips-for-creating-your-plos-figures-graphics/" https://blogs.plos.org/plos/2019/06/looking-good-tips-for-creating-your-plos-figures-graphics/

Reviewers' comments:

Reviewer's Responses to Questions

**Comments to the Author**

1. Is the manuscript technically sound, and do the data support the conclusions?

Reviewer #1: Partly

Reviewer #2: Yes

Reviewer #3: Partly

2. Has the statistical analysis been performed appropriately and rigorously? 

Reviewer #1: Yes

Reviewer #2: Yes

Reviewer #3: Yes

3. Have the authors made all data underlying the findings in their manuscript fully available?

Reviewer #1: No

Reviewer #2: Yes

Reviewer #3: Yes

4. Is the manuscript presented in an intelligible fashion and written in standard English?

Reviewer #1: Yes

Reviewer #2: Yes

Reviewer #3: No

5. Review Comments to the Author

Reviewer #1: The authors looked into the relationship between fear of falling and several multi-domain parameters in Palestinian older adults. The manuscript is generally well written. The authors are to be commended for that. Though the topic is not novel, it could be potentially interesting for the readers of PLOS ONE. However, some issues are to be addressed before the manuscript can be accepted for publication.

Abstract

1. “Fear of falling is a major health problem....”. This is an exaggeration. FOF has serious implications for health, which can be major health problems, but that’s just an indirect relationship. Please rephrase.

2. Several scientific journals from the field or gerontology and geriatrics either advice again or even prohibit using the word “elderly” in submitted manuscripts, as this is a pejorative term. Please use “older adults” or other terms instead throughout the manuscript.

3. “The Falls Efficacy Scale-International in order to predict falls among Palestinian older adults”: The sentence is incomplete. Please rephrase.

4. At the results section the Timed Up-and-Go test is mentioned, although it wasn’t listed in the methods section. (Note that the TUG test does not fall under the category “ADL or IADL)

Introduction

5. “occasional fall”: Do you mean maybe “accidental fall”?

6. “… a ground of high rates…”: what does this mean? Please rephrase

7. There are plenty of previous studies assessing the relationships between fear of falling and various multidisciplinary factors (partly the exact same with the ones used in this study); many of them even using a more robust design (longitudinal studies)*, which is more appropriate to define an outcome`s predictors. What is the exact research gap the authors try to address with this study? Does it have to do with the ethinicity of this sample? If so, how are Palestinians different from the samples of previous studies? Are different results expected?

*e.g:

Austin, N., Devine, A., Dick, I., Prince, R., & Bruce, D. (2007). Fear of falling in older women: a longitudinal study of incidence, persistence, and predictors. Journal of the American Geriatrics Society, 55(10), 1598-1603.

Rivasi, G., Kenny, R. A., Ungar, A., & Romero-Ortuno, R. (2020). Predictors of incident fear of falling in community-dwelling older adults. Journal of the American Medical Directors Association, 21(5), 615-620.

8. What is the reason for choosing the selected parameters as predictors of FOF in this study?

Methods

9. How can the authors be sure that this was a “representative” sample of Palestinian older adults?

10. Why wasn’t the sample size calculation based on the regression analysis, which was also the main analysis of the study?

11. The last paragraph of the “sample size calculation and sampling method” is a bit confusing (mostly linguistically). Please rephrase.

12. Why were nutritionists selected as an appropriate profession to conduct motor functioning geriatric assessments?

13. There are several typos/language errors throughout the manuscript. Please make a careful check before re-submission.

14. What was the rationale for building two categories of people based on their sleeping duration/problems? What does that have to do with FOF and the overall aims of the study and analysis? And what is the classification based on? Please explain and also provide references.

15. “Many screening instruments are available to assess concerns about falling among elderly people”: I am not familiar with any other besides the FES. Can you please name some?

16. The section “nutritional status” barely contains anything related to nutrition. A title like “Body Composition” seems more appropriate.

17. “Cognitive function was assessed using a validated Arabic version of the short geriatric depression scale (GDS-15) in order to detect depression among older adults”, and “The presence of depressive symptoms was assessed using a validated Arabic version of Montreal cognitive assessment –basic to detect mild cognitive impairment.”: Depression and cognition may be often related but there are completely different constructs. Please explain what you mean.

18. ADL and iADL are highly correlated. Why use both as possible predictors and have you checked for multicollinearity?

Results

19. Figures 1 and 2 are a too simplistic and in my opinion unnecessary. The information presented there can be easily integrated into the text (which was partly already done-in figure 2)).

20. Please discuss the unbalance between the number of men and women in the sample and how this can affect the results.

21. The descriptive results take up most of the results section. I suggest to shorten and focus on the associations.

22. “Table 3 shows the relationship between…..”: This table presents descriptive results (means). In what way can relationships be derived from this table?

Discussion

23. Why was such focus given on nutrition (in the discussion but also throughout the manuscript) when nutrition is just one of the many factors that potentially relate to FOF? Following up on this point, why was nutrition even selected as a factor? Please explain the direct (if it exists) relationship between nutrition and FOF.

Reviewer #2: I have reviewed the manuscript entitled Predictors of fall risk among community-dwelling older adults: cross-sectional study from Palestine. Please see my comments below.

Please use the term older adults instead of elderly

Abstract

Present first the assessment used for fall, as it is the main variable of the study.

Present the % of men, women

Present statistical results to support the findings

Conclusion, does not make the responsibility for healthcare professionals and yes tell about policies and procedures. Because this is a correlational study, conclude using this aspect and not about interventions.

Introduction

The introduction presents important information. Because this could be one of the first study in Palestine (it is not possible to confirm as not all studies are available online and in databases), the authors could explain what the particular relevance of this study for the Palestinian population and healthcare are. Also, do not forget to identify the missing in the literature this study wants to cover and the state the study-question. Do not forget to link the conclusion section with the study question.

Methods (please use Methods and not Methodology)

Report fear of falling first

For the heading of Cognitive function, change for Depressive Symptoms and Cognitive function as you are presenting GDS and MOCA.

Statistical analysis section needs to be expanded, giving more details, in special about the regression analysis. This should be as a receipt, telling step by step, you can refer the Tables and figures in the results.

Results

All the topics about patients’ recruitment should appear in the Methods section and not Results.

Tables need to be formatted.

Reviewer #3: Thank you for inviting me to review this manuscript. This study is about falls risk predictors among community-dwelling older adults in Palestine. Although well-intended, the study has shortcomings that limit my enthusiasm.

A first comment is for the (in)congruence between the title – “Predictors of fall risk…” and the aims of the study - “… to investigate risk for fear of falling among Palestinian older adults….” . Risk of falls and risk for fear of falling are conceptually different. Fear of falling can be a determinant of risk of falls and also a consequence of falls. So, the authors need to be clearer about the differences. In fact, when I read the manuscript, it seems that the authors intended to explore factors related to fear of falling, rather than to risk of falling. In this sense, please elaborate more on the relation about predictors of fear of falling -fear of falling – and risk of falling.

Moreover, predictors of falls or predictors of fear of falling are well established in the literature, thus I cannot see what the novelty of this study is, beside cultural issues, like being conducted among Palestinian older population. If this is the case, the authors should address how cultural issues might have impacted the results in the discussion section.

In the Introduction, the background needs to be updated regarding epidemiological data about the world aging population, as there are recent references about demographic projections besides that of Newgarde et al. 2013. Since FES-I is the instrument used to assess the variable of interest in this study, the authors also need to revise the different constructs of fear associated with falls and revise which construct they really are aiming to assess – Concern about falling. For inspiration read this article:

J Aging Health. 2011 Feb;23(1):3-23. doi: 10.1177/0898264310378039. Epub 2010 Sep 17.

Reconceptualizing the role of fear of falling and balance confidence in fall risk.

Hadjistavropoulos T1, Delbaere K, Fitzgerald TD.

A last comment regarding the background is related to the 4th paragraph. A more clear and sustained relation about the fear of falling and variables such as mobility, functional decline, depression, activity restriction, just to name a few, should be emphasised.

The methods section is not presented in a standard way: Study design, Participants, Data collection; data analysis.

Regarding exclusion criteria, how was “having dementia” assessed? Were participants allowed to use walking aid?

Did participants report a history of falls in the last 3, 6 or 12 months?

Please provide cut-off scores for the IPAQ and interpretations.

Please go into more detail regarding the psychometric properties of the instruments used (e.g., FES-I, MOCA, GDS, TUG, etc.)

GDS does not assess cognitive function but depressive symptoms. Please correct this information for DGS and MOCA

In the results section, there are too many tables and figures that are not easy to follow.

Please provide a table describing the sample’s sociodemographic, without separating women and man. It is not clear why the authors present data for men and women separately without any statistical analysis between them in Table 1. Finally, all the English must be carefully revised.

6. PLOS authors have the option to publish the peer review history of their article (what does this mean?). If published, this will include your full peer review and any attached files.

Reviewer #1: **Yes: **Eleftheria Giannouli

Reviewer #2: **Yes: **Allan Bregola

Reviewer #3: No

---

## [Author Response · Author response to Decision Letter 0]

17 Jun 2022

the responses is uploaded as a file named (response to reviewers)

---

## [Decision Letter · Decision Letter 1]

26 Sep 2022

PONE-D-21-23236R1Predictors of fear of falling among community-dwelling older adults: cross-sectional study from Palestine.PLOS ONE

Dear Dr. Badrasawi,

Thank you for submitting your manuscript to PLOS ONE. After careful consideration, we feel that it has merit but does not fully meet PLOS ONE’s publication criteria as it currently stands. Therefore, we invite you to submit a revised version of the manuscript that addresses the points raised during the review process.

ACADEMIC EDITOR: You did a good job on responding to the previous reviewers comments and suggestions. However, since the previous reviewers were not responsive, we had to send it to other reviewers. Please, revise according to their suggestions and submit it again. 

We look forward to receiving your revised manuscript.

Kind regards,

Aqeel M Alenazi

Academic Editor

PLOS ONE

Journal Requirements:

Additional Editor Comments:

You did a good job on responding to the previous reviewers comments and suggestions. However, since the previous reviewers were not responsive, we had to send it to other reviewers. Please, revise according to their suggestions and submit it again.

Reviewers' comments:

Reviewer's Responses to Questions

**Comments to the Author**

1. If the authors have adequately addressed your comments raised in a previous round of review and you feel that this manuscript is now acceptable for publication, you may indicate that here to bypass the “Comments to the Author” section, enter your conflict of interest statement in the “Confidential to Editor” section, and submit your "Accept" recommendation.

Reviewer #4: All comments have been addressed

Reviewer #5: All comments have been addressed

Reviewer #6: (No Response)

2. Is the manuscript technically sound, and do the data support the conclusions?

Reviewer #4: Yes

Reviewer #5: Yes

Reviewer #6: Yes

3. Has the statistical analysis been performed appropriately and rigorously? 

Reviewer #4: Yes

Reviewer #5: Yes

Reviewer #6: Yes

4. Have the authors made all data underlying the findings in their manuscript fully available?

Reviewer #4: Yes

Reviewer #5: Yes

Reviewer #6: Yes

5. Is the manuscript presented in an intelligible fashion and written in standard English?

Reviewer #4: Yes

Reviewer #5: Yes

Reviewer #6: Yes

6. Review Comments to the Author

Reviewer #4: Fear of falling and its predictors among older adults is an important topic. Exploring those factors in specific community would help health care providers detect fear of falling early on and address them before getting serious injuries.

Authors of this manuscript did a great job in investigating several important factors and analyze the results in the appropriate statistics.

Reviewer #5: 1. Under Sample size and sampling techniques: a small country divided into three regions (south, center, and south) need to correct it.

2. Reference: there is differences in the reference style

Reviewer #6: Overall, this is a well-written manuscript of a well-designed study. I am highlighting the areas in which I believe the manuscript could be clarified or improved.

Abstract:

1. For abbreviations and acronyms spell out the full term at its first mention, like ADL, IADL

Introduction:

-Please add appropriate reference in the first paragraph

- I would suggest adding a paragraph talking about falls and fear of falling in the Middle east, and emphasize how serious and prevalent is fear of falling. ( suggested articles DOI: 10.1016/j.archger.2019.04.006, DOI: 10.1111/ajag.12673 , DOI: 10.1016/j.archger.2018.08.001

Methods:

- In the method section, the author should report whether there is any data missing and how missing data is processed. Also, add a flow chart of the participants selection.

- Use “and” instead of using “&”

- History of falls has been consistently reported as a risk factor of fear of falling in previous studies. Have taken it into consideration if not, why?

- In your inclusion criteria of age was 60 years and older, and yet in your sample the age ranges from 65-98. Please explain?

7. PLOS authors have the option to publish the peer review history of their article (what does this mean?). If published, this will include your full peer review and any attached files.

Reviewer #4: **Yes: **Alia Alghwiri

Reviewer #5: **Yes: **Bijad Alqahtani

Reviewer #6: No

---

## [Author Response · Author response to Decision Letter 1]

7 Oct 2022

Comment Reply Changes on the manuscript 

Editor comments You did a good job on responding to the previous reviewers comments and suggestions. However, since the previous reviewers were not responsive, we had to send it to other reviewers. Please, revise according to their suggestions and submit it again. Thank you 

 Please review your reference list to ensure that it is complete and correct Done Please refer to the manuscript file 

Reviewer #6 Overall, this is a well-written manuscript of a well-designed study. I am highlighting the areas in which I believe the manuscript could be clarified or improved. 

 Abstract: For abbreviations and acronyms spell out the full term at its first mention, like ADL, IADL 

 Corrected 

Added as full term in the first time and functional status (using activities of daily living and instrumental activities of daily living scale),

 Introduction:

 -Please add appropriate reference in the first paragraph

- I would suggest adding a paragraph talking about falls and fear of falling in the Middle east, and emphasize how serious and prevalent is fear of falling. ( suggested articles DOI: 10.1016/j.archger.2019.04.006, DOI: 10.1111/ajag.12673 , DOI: 10.1016/j.archger.2018.08.001 

 Done

Thank you, the references were added 

 In general, this elevation in the relative percentage in comparison to the overall population has great implications both for health services and for older adults [1].

In a systematic review, included six studies conducted among Arabic population, Gulf Cooperation Council Countries, the pooled prevalence of fall was 46.9%ranged from 34% to 57.7% [9]. While in Turkey, the prevalence of falls among older adults aged 80 years and above was 35.4%, and fear of falling was 86.6% [10]. Among Iranian population, the prevalence of falls was 39.7% among 653 older adults [11].

The references were also added to the references list

 Method 

In the method section, the author should report whether there is any data missing and how missing data is processed. Also, add a flow chart of the participants selection. 

Use “and” instead of using “&”

History of falls has been consistently reported as a risk factor of fear of falling in previous studies. Have taken it into consideration if not, why?

In your inclusion criteria of age was 60 years and older, and yet in your sample the age ranges from 65-98. Please explain? 

There were 10 participants who were excluded due to missing data 

We added the flow chart in the primary submission 

Please see attached figure below the table 

Corrected 

We agree with this valuable comment. We have the data for history of fall and we are currently writing a manuscript having the history of fall as main variable.

The inclusion criteria was 65 

We correct it 

Please refer to the first submission 

Please refer to the manuscript 

No change in the manuscript 

In the current study, participants were included if they aged 65 years old or above, and willing to participate and to provide all the required data.

Reviewer #4 

 Fear of falling and its predictors among older adults is an important topic. Exploring those factors in specific community would help health care providers detect fear of falling early on and address them before getting serious injuries.

Authors of this manuscript did a great job in investigating several important factors and analyze the results in the appropriate statistics.

Thank you 

Reviewer #5: 

 1. Under Sample size and sampling techniques: a small country divided into three regions (south, center, and south) need to correct it.

2. Reference: there is differences in the reference style

Corrected

All the reference were revised and corrected 

(north, center, and south)

---

## [Decision Letter · Decision Letter 2]

18 Oct 2022

Predictors of fear of falling among community-dwelling older adults: cross-sectional study from Palestine.

PONE-D-21-23236R2

Dear Dr. Badrasawi,

We’re pleased to inform you that your manuscript has been judged scientifically suitable for publication and will be formally accepted for publication once it meets all outstanding technical requirements.

Kind regards,

Aqeel M Alenazi

Academic Editor

PLOS ONE

Additional Editor Comments (optional):

Thank you for addressing the comments from all reviewers. We apologize for unexpected delay with your manuscript. You did a great Job in responding to reviewers

Reviewers' comments:

Reviewer's Responses to Questions

**Comments to the Author**

1. If the authors have adequately addressed your comments raised in a previous round of review and you feel that this manuscript is now acceptable for publication, you may indicate that here to bypass the “Comments to the Author” section, enter your conflict of interest statement in the “Confidential to Editor” section, and submit your "Accept" recommendation.

Reviewer #5: (No Response)

Reviewer #6: All comments have been addressed

2. Is the manuscript technically sound, and do the data support the conclusions?

Reviewer #5: (No Response)

Reviewer #6: Yes

3. Has the statistical analysis been performed appropriately and rigorously? 

Reviewer #5: (No Response)

Reviewer #6: Yes

4. Have the authors made all data underlying the findings in their manuscript fully available?

Reviewer #5: (No Response)

Reviewer #6: Yes

5. Is the manuscript presented in an intelligible fashion and written in standard English?

Reviewer #5: (No Response)

Reviewer #6: Yes

6. Review Comments to the Author

Reviewer #5: (No Response)

Reviewer #6: Overall, this is a well-written manuscript of a well-designed study. Thank you for addressing my comments.

7. PLOS authors have the option to publish the peer review history of their article (what does this mean?). If published, this will include your full peer review and any attached files.

Reviewer #5: No

Reviewer #6: No

---

## [Editor Report · Acceptance letter]

19 Oct 2022

PONE-D-21-23236R2 

Predictors of fear of falling among community-dwelling older adults: cross-sectional study from Palestine. 

Dear Dr. Badrasawi:

I'm pleased to inform you that your manuscript has been deemed suitable for publication in PLOS ONE. Congratulations! Your manuscript is now with our production department. 

Kind regards, 

on behalf of

Dr. Aqeel M Alenazi 

Academic Editor

PLOS ONE